# Analysis of High Temporal Resolution Land Use/Land Cover Trajectories

**Jean-François Mas [1,2,\*]** , **Rodrigo Nogueira de Vasconcelos [2,3]** and **Washington Franca-Rocha [2,3]**

1   Centro de Investigaciones en Geografía Ambiental, Universidad Nacional Autónoma de México, CP 58190 Morelia, Michoacan, Mexico
2   Instituto Nacional de Ciência e Tecnologia em Estudos Interdisciplinares e Transdisciplinares em Ecologia e Evolução (INCT IN-TREE), CEP 40110-909 Salvador, Bahia, Brazil; rnvuefsppgm@gmail.com (R.N.d.V.); wrocha@uefs.br (W.F.-R.)
3   PPG em Modelagem e Ciências da Terra e do Ambiente, Universidade Estadual de Feira de Santana, CEP 44036-900 Feira de Santana, Bahia, Brazil
\*   Correspondence: jfmas@ciga.unam.mx; Tel.: +52-443-322-3835

**Abstract:** In this study, methods, originally developed to assess life course trajectories, are explored in order to evaluate land change through the analysis of sequences of land use/cover. Annual land cover maps which describe land use/land cover change for the 1985–2017 period for a large region in Northeast Brazil were analyzed. The most frequent sequences, the entropy and the turbulence of the land trajectories, and the average time of permanence were computed. Clusters of similar sequences were determined using different dissimilarity measures. The effect of some covariates such as slope and distance from roads on land trajectories was also evaluated. The obtained results show the potential of these techniques to analyze land cover sequences since the availability of multidate land cover data with both, high temporal and thematic resolutions, is continuously increasing and poses significant challenges to data analysis.

**Keywords:** land use/cover change; deforestation; change trajectory; sequences; representative patterns; caatinga

---

## 1. Introduction

Land use/cover changes (LUCC) is the focus of a large number of studies due to its relationship with global and regional change processes such as biodiversity loss, climate change, erosion, flooding, etc. A sound evaluation of the temporal dynamics of LUCC is needed to understand its effects on ecosystems. Watson et al. [1] identified four key components of land-change regimes: (i) Frequency of LUCC over a period of time, (ii) the sequence of land-cover types, (iii) the time period over which each land cover class remains, and (iv) the magnitude of the difference between the land cover classes.

Remote sensing has proven to be the most cost-effective method to monitor LUCC over large territories. Landscape dynamics can be assessed and analyzed by using multidate satellite imagery and geographic information system (GIS) techniques. Typically, multidate data consists of images acquired on the same geographical area at a few acquisition dates over a rather large period. For instance, Zaehringer et al. [2] used Landsat imagery to assess LUCC for two intervals, 1995–2005 and 2005–2011 in northeastern Madagascar. The most common approaches used to describe the changes are the computing of changes rates (e.g., rate of deforestation), the elaboration of change and Markov matrices, and metrics as persistence, gains, losses, swaps and change density [3–6]. However, these approaches are based on the comparison of only two dates and are repeated for each period in order to depicts changes in the LUCC patterns over the entire extent of time.

Recently, a few authors analyzed the chronological succession of land use/cover over time. However, in order to reduce the number of combinations, they used a small number of categories and time steps. For instance, Baral et al. [4] depicted eight forest cover change trajectories using two categories (forest/non forest) and three dates. Zhang et al. [7] classified 61 wetland land cover change trajectories, based on three land cover classes and five dates, using similarity, turnover, and diversity indices. Nevertheless, nowadays the increased number of satellites with improved temporal resolution allows the acquisition of frequent images and cloud computing enables users to process huge multidate data sets in order to produce LUCC maps with a high temporal resolution. For example, the global Forest Cover database describes forest gain and loss at 30 m resolution, for the entire globe on an annual basis during 2000–2014 [8]. During the last years, several time series of land use and land cover maps have been elaborated at global scale (e.g., MODIS MCD12 product, Climate Change Initiative (CCI) Land Cover Annual Global Land Cover Maps) or for different regions of the world [9–16]. These increasingly available LUCC data with high temporal and thematic resolutions pose the challenge of developing novel methods able to process high volumes of data which describe the succession of land use/cover types over time [15].

Otherwise, the analysis of sequences is common in several disciplines including genetics (e.g., DNA sequences) [17,18], marketing (analysis of shopping habits, visits to web pages) [19], clinical studies (e.g., analysis of progressive diseases) [20–22], and social sciences (life course trajectories) [23,24]. In these disciplines, methods and tools for sequence analysis have been developed. For example, the description of human behavior over time is central to many problems treated in the social sciences arena. The life course trajectories are sequences of events that individual present over time and that describe processes as the transition to adulthood, education, professional careers, the constitution of family, or criminal activities. These trajectories seek to represent the dynamics of people's experiences, through the different episodes that they experience throughout their lives, and can be analyzed qualitatively, quantitatively or both [25]. Quantitative analysis of these sequences, based on measures of dissimilarity between individual trajectories, aim at identifying typical pathways and associating them to different covariates. The objective of this current study is to evaluate the application of statistical techniques used in life course trajectory analysis to temporal sequences of land use/cover in order to identify potentialities and limitations of this approach.

The article is organized as follows: In Section 2, life course trajectories analysis and some data-mining-based methods are presented, and their potential application to LUCC trajectories is discussed. Then, an illustrative application, based on multidate data from the MapBiomas project [26], is presented. The study area, the materials, and the methods are described in Sections 3–5. In Section 6, the results of the case study are presented. Finally, a few concluding remarks are made in Section 7.

## 2. Life Course Trajectories Analysis and Its Potential Application to LUCC Trajectories

Abbott [27] first emphasized the importance of the analysis of the sequence of social events. Nowadays, sequence analysis is a method commonly used to study life and career trajectories because it allows the determination of trajectory patterns to provide a holistic view taking into account all the states experienced. In comparison, survival analyses, commonly used in biomedical research and reliability research, focus on the timing of only one specific event, and do not give a general picture of the trajectories.

In social sciences, the sequence exploration can concern state or event sequences. A state, such as "unemployed", lasts the entire unit of time while an event, as "finding a job", occurs at a particular time point. The event has no duration, and provokes, likely in conjunction with other events, a state change. In the state sequences, the position in the sequence indicates the duration since the commencement of the sequence. In the event sequences, the position only gives the number of precedent events. Thus, state sequences mainly aim at studying durations and timing in life trajectories, while event sequences focus on the order in which events occur. Another fundamental difference between states and events is that various events can occur at the same time, while states are mutually exclusive [28]. State sequences

have received more attention than event sequences in the social sciences literature, particularly since the implementation of methods aimed at processing state sequences data sets [29]. In LUCC study, both approaches can be envisaged. Events can be, for instance, a land cover conversion, a wildfire or a drought. However, it can be more straightforward to derive state sequences from a stack of multidate maps or classified images and consider a land use/cover category as a state. This paper will focus on state sequences rather than event sequences.

Sequence analysis can present different facets: exploratory (looking for patterns), and explanatory (analyzing why some patterns are related to certain conditions). Studer and Ritschard [30] identified several important aspects of the information contained in life course trajectories' state sequences:

- The distinct observed states present in a set of sequences,
- The within-sequence state distribution,
- The timing (the date at which each state occurs),
- The duration (the consecutive and total time spent in the different successive states) and,
- The sequencing (the order of the different successive states).

For instance, Figure 1 shows five sequences with three different states (A, B and C) along seven time steps. In LUCC studies, spatial distribution of the different sequence patterns is an additional dimension.

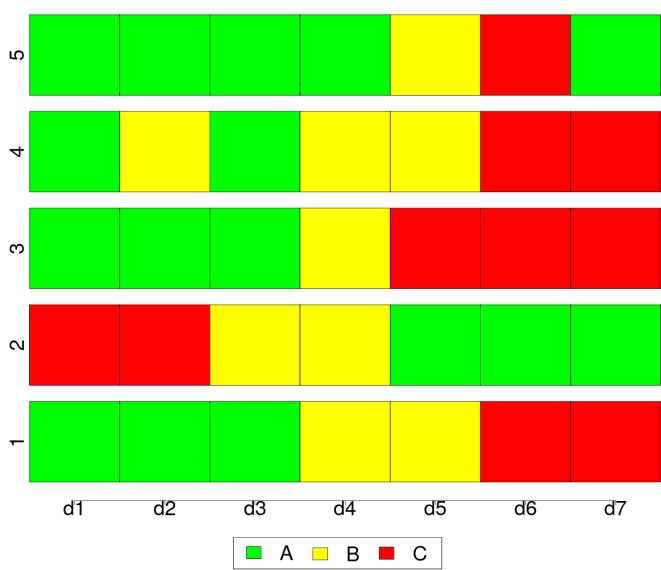

**Figure 1.** Sequences of five individuals/sites.

Longitudinal and transversal patterns take into account the individual trajectories and all the states observed at a certain time respectively. Therefore, a large number of different possible patterns can be expected depending on the number of possibles states and of time steps. Generally, the first step in sequence analysis is the graphical representation of all individual sequences, the computing of the average time spent in each state, the rates of transitions between states and metrics that describe state distributions and complexity. For instance, the entropy uses the number of different states as a measure of complexity. In LUCC, this calculation can be done using the states observed at a certain time (transversal entropy) to describe landscape complexity or using the states observed for the same place (longitudinal entropy) over time.

The exploration of sequences aims at finding the most typical sequence, which is defined as the succession of states that best represents the entire group [31]. The identification of typical sequence was often done by visual inspection [32,33]. However, this approach is subjective and not feasible when the number and the diversity of sequences increases. For these reasons, several data driven approaches

have been proposed. For instance, Aassve et al. [31] proposed (i) the modal state sequence which is the sequence obtained by taking at each position the modal state at that position and, (ii) the "medoid", the sequence that minimizes some function of the distances to all other sequences. It is worth noting that the modal approach can be inconsistent because it does not necessarily give an observed sequence as result. Advanced sequence analyses are based on the pairwise distance, or dissimilarity, between sequences. For example, Gabadinho and Ritschard [34] proposed a method to determine a subset of the observed sequences able to represent the entire set of sequences. This method uses pairwise dissimilarities between sequences to identify the principal types of patterns by clustering the sequences using these measures. The evaluation of the dissimilarity between sequences is the most common starting point for dissimilarity-based sequence methods. There are many dissimilarity measures, and some of them are presented in the method section.

Sequence analysis can also aim at assessing the relationship between sequencing of states/events and covariates. For instance, in social sciences, it can be interesting to compare groups of sequences that belong to different social groups or to determine the sub-sequences that best discriminate groups or depend on a covariate value. In LUCC studies, such approaches can be useful in identifying different processes of change and assessing the role of drivers. A R script file along with example data and a guide are available at Supplementary Material.

## 3. Study Area

The case study was conducted in the Northeastern Region of Brazil. The area is located between 38°17′06″ W and 44°56′02″ W and between 9°39′11″ S and 14°04′26″ S and has a total area of 340,500 km². It encompasses a large portion of the State of Bahia and a small part of Piauí (Figure 2). It comprises three biomes: Cerrado, Caatinga and Mata Atlântica.

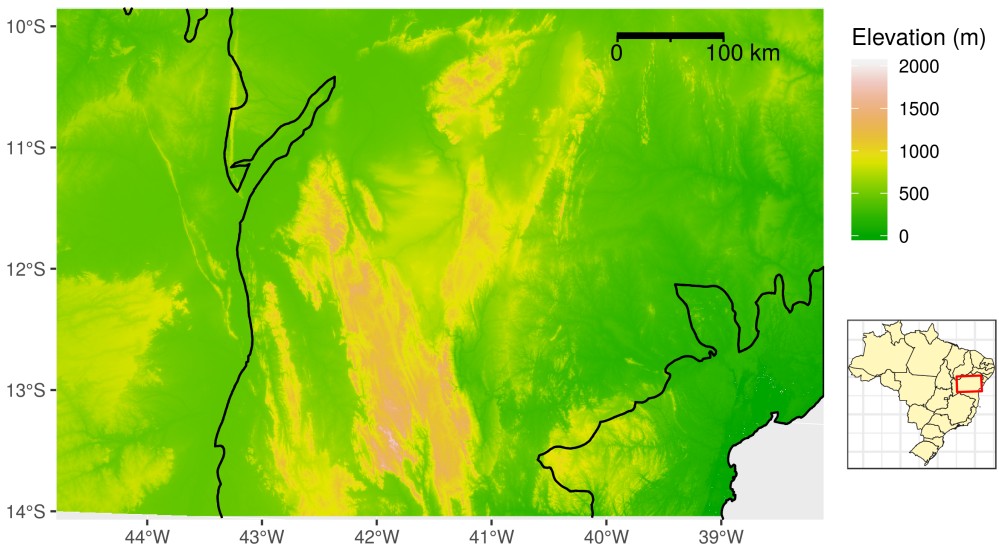

**Figure 2.** Location and elevation data of the study area. The black lines indicate the distribution of the biomes. From west to east: Cerrado, Caatinga and Mata Atlântica. Inset: Study area location in Brazil (red rectangle).

## 4. Materials

The MapBiomas project (http://mapbiomas.org), a network of collaborative institutions derived from the System for Estimation of Green House Gases Emissions (SEEG) initiative, maps the main types of land use /cover annually between 1985 and 2017 for the principal Brazilian biomes in order to contribute to the understanding of the dynamics of land use in Brazil [26]. These annual time series of land cover maps (collection 3) were obtained at 30 m resolution through the classification of Landsat

imagery using Random Forest in the Google Earth Engine cloud. The classification scheme comprises 20 categories for entirety of Brazil (Table 1). Spatial and temporal filters were applied to the classified images to eliminate the salt and pepper effect and reduce unlikely transitions. A digital elevation model (30 m resolution) from the Advanced Land Observing Satellite (ALOS) Global Digital Surface Model (https://www.eorc.jaxa.jp/ALOS/), annual precipitation and precipitation of driest quarter maps (WorldClim bioclimatic variables, 1km resolution, http://www.worldclim.org/bioclim) and a digital map of roads from the Brazilian Institute of Geography and Statistics (IBGE), the Brazilian mapping agency, were also used. The analyzes were carried out using the R platform [35], in particular the packages fastcluster [36], raster [37] and TraMineR [38].

**Table 1.** Classification scheme of MapBiomas cartography (collection 3).

| Level 1 | Level 2 | Level 3 |
| --- | --- | --- |
| 1. Forest | 1.1. Natural forest | 1.1.1. Forest formation |
|  |  | 1.1.2. Savanna formation |
|  |  | 1.1.3. Mangrove |
|  | 1.2. Forest plantation |  |
| 2. Non forest natural formation | 2.1. Wetland |  |
|  | 2.2. Grassland formation |  |
|  | 2.3. Salt flat |  |
|  | 2.3. Other non forest natural formation |  |
| 3. Farming | 3.1. Pasture | 3.1.1. Natural Pasture |
|  |  | 3.1.2. Planted Pasture |
|  | 3.2. Agriculture |  |
|  | 3.3. Mosaic of agriculture and pasture |  |
| 4. Non vegetated area | 4.1. Beach and dune |  |
|  | 4.2. Urban infrastructure |  |
|  | 4.3. Rocky outcrop |  |
|  | 4.4. Mining |  |
|  | 4.5. Other non vegetated area |  |
| 5. Water | 5.1. River, lake and ocean |  |
|  | 5.2. Aquaculture |  |
| 6. Non observed |  |  |

## 5. Methods

### 5.1. Preprocessing

The MapBiomas maps were reclassified to handle a smaller number of classes: (1) Forest, (2) Savanna, (3) Grasslands, (4) Mosaic of agriculture and pastures and (5) others (regrouping urban areas, areas without vegetation, mangroves and water bodies). Then a temporal filter was applied in order to reduce the size of the temporal sequences and the amount of unlikely transitions: The entire period 1985–2017 was divided into 11 three year periods, represented by its modal category. In the case of three different categories occurring during the three year period, no unique modal category could be found and the modal category of the following three year period was selected (for the last period, the modal category from the previous period was used). Digital elevation tiles were mosaicked and a slope map was produced. Euclidean distance from roads was also computed. A raster database, resampled to 500 m using the nearest neighbor algorithm, was elaborated to integrate the land use/cover maps with the covariates (elevation, slope, climatic variables and distance maps). The resampling from 30 m to 500 m enabled us to reduce the huge amount of data drastically, without loss of information (diversity of sequences) and spatial coherence when mapping the results. Pixel values were extracted to generate tabular data which consist of a temporal sequence of land use/cover ("states"), complemented with the covariates.

### 5.2. Land Use/Cover Sequence Analysis

Graphs of the observed sequences were elaborated and a descriptive analysis was carried out: The most frequent sequences were determined, both space and temporal complexities were evaluated

by computing the transversal and longitudinal entropy index (also known as the Shannon index) (Equation (1)). The rationale of this index is that the more different states occur in the sequence, and the more equal their respective proportion, the more difficult it is to accurately predict which state will be the next one in the sequence. Thus, the entropy quantifies the uncertainty ("degree of surprise") associated with this prediction. As the entropy does not take into account the states' ordering, Elzinga and Liefbroer [39] proposed another index, called the sequence turbulence. The turbulence is based on the number of different subsequences in the sequence (Equation (2)). A sequence s is a subsequence of S when all the successive elements $s_i$ of s appear in S in the same order. It is worth noting that unshared states can occur between those common to both sequences s and S. For instance, s = A-A-C is a subsequence of S = A-A-A-B-B-C-C [38].

$$E = - \sum_{i=1}^{s} p_i \log(p_i) \tag{1}$$

where $E$ is the entropy, $p_i$ is the proportion of category $i$ and $s$ the total number of categories. Note that, additionally, the entropy is normalized by dividing the obtained value by the theoretical maximum.

$$T = log_2 \left( \phi(x) \frac{s_{t,max}^2(x) + 1}{s_t^2(x) + 1} \right) \tag{2}$$

where $T$ is the turbulence, $\phi(x)$ the number of distinct subsequences, $s_t^2(x)$ the variance of the consecutive times $t_j$ spent in the distinct states and $s_{t,max}^2(x)$ is the maximum value that $s_t^2(x)$ can take given the total duration of the sequence.

*5.3. Pairwise Dissimilarities between Sequences*

The dissimilarity between sequences was assessed using different measures: The longest common subsequence (LCS) distance, the longest common prefix (LCP) and the optimal matching (OM).

The longest common subsequence (LCS) distance between two sequences x and y is based on the length $A(x, y)$ of the longest common subsequence and is formally defined as [38,39]:

$$\delta(x, y) = A(x, x) + A(y, y) - 2A(x, y) \tag{3}$$

where $A(x, x)$ and $A(y, y)$ are the length of $x$ and $y$ respectively and $A(x, y)$ is the length of the longest common subsequence between $x$ and $y$. The less common states between the two sequences, the larger the distance. For instance, the longest common subsequence between sequences 1 and 4 (Figure 1) is A-A-B-B-C-C (length = 6) and thus the distance is 2 ($7 + 7 - 2 \times 6$).

The longest common prefix (LCP) takes into account the first state(s) of the sequences. For instance, the $k$th prefix is defined by the k first states of the sequence. The length of the longest common prefix of two sequences is used to calculate the LCP distance $\delta_p(x, y)$ [38,39]:

$$\delta_p(x, y) = |x| + |y| - 2A_p(x, y) \tag{4}$$

where $|x|$ and $|y|$ are the length of sequences x and y and $A_p(x, y)$ is the length of the longest common prefix between sequences $x$ and $y$. For example, the LCP distance between sequences 2 and 5 (Figure 1) is 14, the maximum possible distance, because the first state of the two sequences is different and thus, they do not have the same prefix. The longest common prefix between sequences 1 and 4 is A and therefore the distance is 12 ($7 + 7 - 2 \times 1$).

The optimal matching (OM) distance $\delta(x, y)$ between x and y is defined as the cost of transforming $x$ into $y$ using insertions, deletions and substitutions of states. Thus, the value of the distance $\delta(x, y)$ depends on the substitution and insertion or deletion (indel) costs. The indel cost is mostly set as a constant while the substitution costs frequently depend on the concerned states. For example, sequence 5 can be obtained from sequence 3 inserting A at the beginning of the sequence, deleting the last C and

substituting C by A in the seventh state. The total cost is the cost of one insertion, one deletion and one substitution. OM is then the sum of two terms: A weighted sum of time shifts represented by indels and a weighted sum of the mismatches, represented by the substitutions which remain after the time shifts. Large indel costs make the dissimilarity measure highly time sensitive, while low indel costs, in comparison to substitution costs, reduce the importance of time shifts in sequence comparisons [30].

Substitution and indel costs are therefore critical to carry out OM. Studer and Ritschard [30] reported three main approaches to choose substitution costs:

- The first approach consists in determining the costs on a theoretical base to evaluate the similarity of two states. For example, in career trajectory, Senior Manager is closer to Manager than to Employee and in order to reflect this hierarchy, the cost of replacing Senior Manager with Employee can be set higher than the cost of substitution between Senior Manager and Manager [30]. In LUCC, a similar hierarchy between categories can be imagined, for instance based on vegetation succession processes.
- Another approach is based on state attributes on which closeness between states is evaluated. For instance, for career trajectories, the qualification required, level of responsibility or the degree of precariousness can be taken into account [30]. In LUCC a similar approach could be envisaged taking into account ecological value associated with each land category.
- A third strategy is to derive the cost from the data. For instance, a common way to obtain the substitution costs is assigning larger costs to substitution between states when the transitions rates are low, and inversely, assigning a smaller cost when frequent transitions are observed. Another approach considers that two states are close when they are frequently followed by a common state.

Indel costs are generally set to a constant value. However, they can be derived from the observed frequency of the state to assign higher costs to rare states [40]. Dissimilarity between sequences was calculated using four methods: LCS, LCP, OM based on the transition rates, and OM based on land cover features. In this last case, an indel cost value broadly associated to biomass was assigned to each category (10 to Forest, 5 to savanna, 3 for pasture, agriculture and mosaic and 1 for others).

A hierarchical cluster analysis was carried out in order to aggregate the similar sequences into a reduced number of groups using Ward's method. Initially, each sequence is assigned to its own cluster and then the algorithm runs iteratively, joining, at each step, the two most similar clusters, until there is only a single cluster [36].

### 5.4. Assessment of the Effect of Covariates

Land change science aims at understanding the role of different drivers in LUCC and explaining the coupled human-environment system dynamics that generate these changes [41,42]. In this study, multi-factor analysis of variance (ANOVA) based on the dissimilarity matrix allows for evaluation of the part of discrepancy explained by covariates. It enables assessing the Type-II effect for each covariate, which is the effect measured when removing the covariate from the complete model with all variables included. The chi-square test was used to identify subsequences that discriminate significantly groups based on covariates. This approach was used to assess the effect of the distance from road on deforestation.

## 6. Results and Discussion

### 6.1. Preprocessing

Figure 3 presents the land use/cover maps for 1985–1987 and 2015–2017. As a broad picture, an expansion of agriculture and pasture areas can be noticed. However, the main distribution of land use/cover categories remains the same between these two dates.The entire area is represented by more than 1,267,500 pixels, and more than 125,800 different sequences were observed. However, 80%

of the area is represented by the 3000 more frequent sequences. Further analyses were based on a random sampling of 46,000 sequences due to the algorithms' processing limitations. Figure 4 presents some randomly selected sequences, where all the sequences are ordered by the initial states and the most common sequences. As it can be observed, the most common sequences are those that present permanence of the same land use/cover throughout the entire period. However, many sequences indicate complex interchanges between savanna, grasslands and mosaics of agriculture and pasture.

**Land use/cover circa 1986**

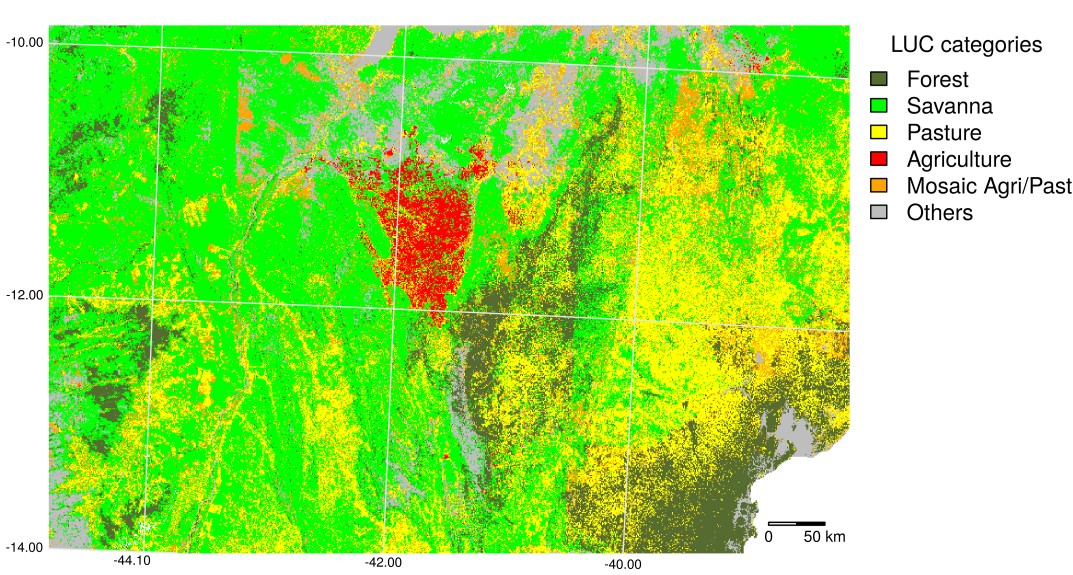

**Land use/cover circa 2016**

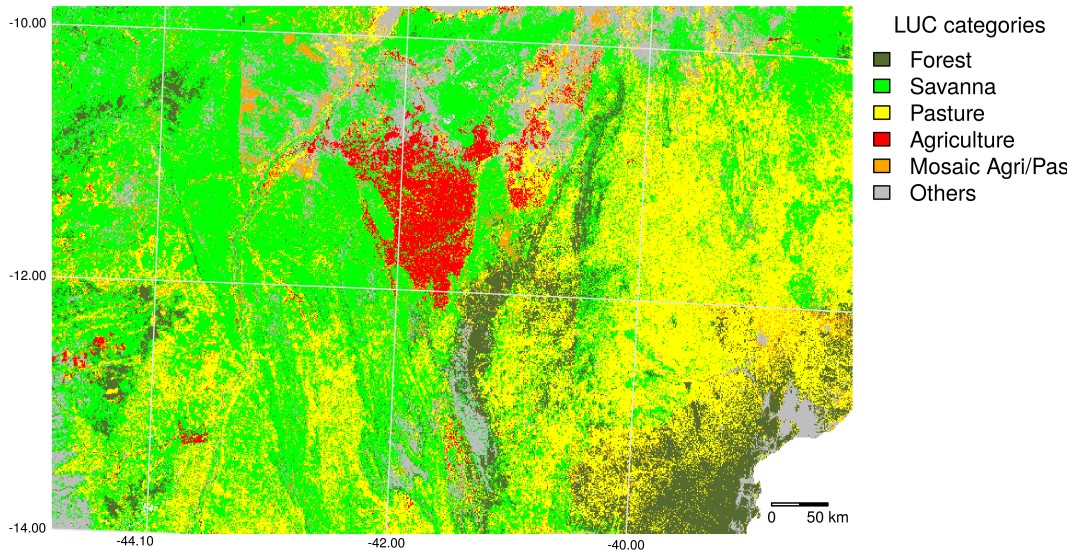

**Figure 3.** Land use/cover (LUC) distribution at the beginning and at the end of the time series.

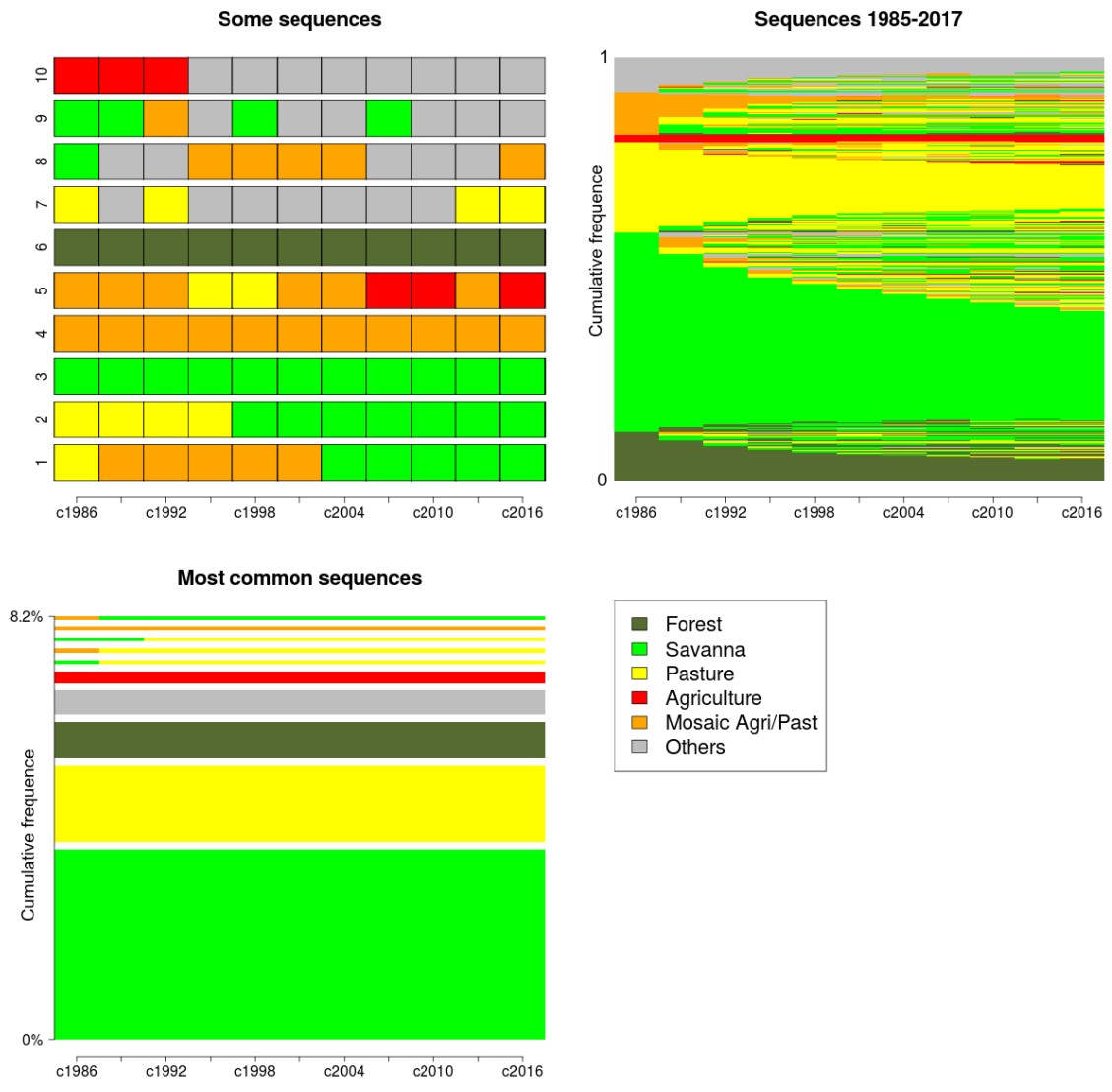

**Figure 4.** Descriptive representation of the land use/cover sequences.

### 6.2. Land Use/Cover Sequence Analysis

Figure 5 indicates the proportion of area covered by the different categories over the period of analysis, the landscape entropy over time, and the average time of permanence of each land cover category. In the left part of the figure, a continuous increase of grassland area, along with a decrease of savanna, forest and mosaic areas, can be observed. However, the proportion of the different types of land cover does not show drastic changes, and, consequently, the entropy index is relatively stable. The study area is dominated by the savanna throughout the studied period. Table 2 shows the rates of transition observed during the entire period. Permanence transitions present rates above 0.8 except for the category *Mosaic* which also presents high rates of transformation into savanna and pasture. Figure 6 shows the distribution of the entropy and turbulence indices, and allows the identification of highly dynamic areas. Both indices, longitudinal entropy and turbulence, represent the complexity of the sequences and were highly correlated (r = 0.93).

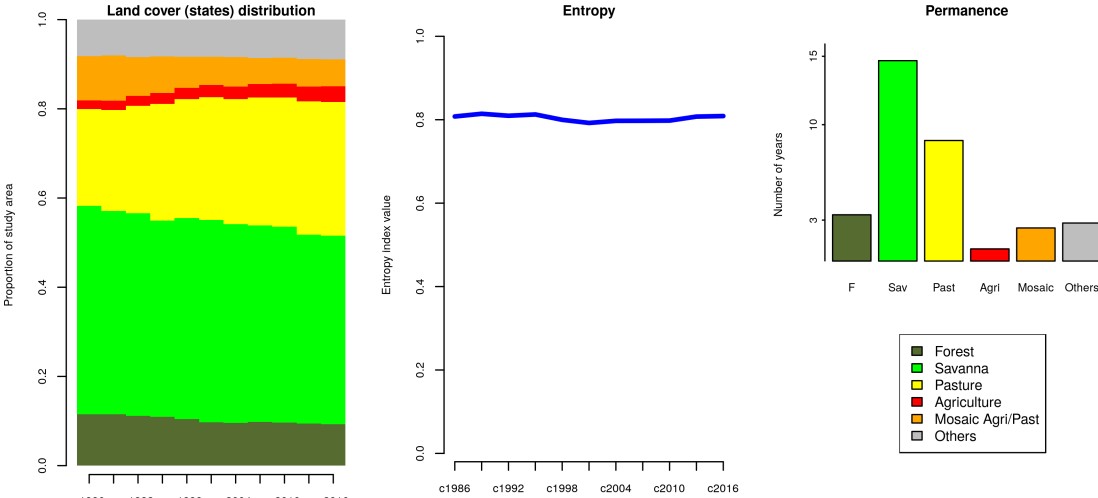

**Figure 5.** Some indexes of the descriptive analysis of sequences.

**Table 2.** Rates of transition between land use/cover categories.

|  | **Forest** | **Savanna** | **Pasture** | **Agriculture** | **Mosaic** | **Others** |
|---|---|---|---|---|---|---|
| Forest | 0.832 | 0.091 | 0.049 | 0.001 | 0.020 | 0.007 |
| Savanna | 0.019 | 0.890 | 0.038 | 0.001 | 0.029 | 0.023 |
| Pasture | 0.016 | 0.046 | 0.884 | 0.007 | 0.040 | 0.007 |
| Agriculture | 0.002 | 0.009 | 0.037 | 0.936 | 0.010 | 0.007 |
| Mosaic | 0.027 | 0.187 | 0.176 | 0.008 | 0.540 | 0.063 |
| Others | 0.007 | 0.110 | 0.029 | 0.003 | 0.052 | 0.798 |

## Entropy

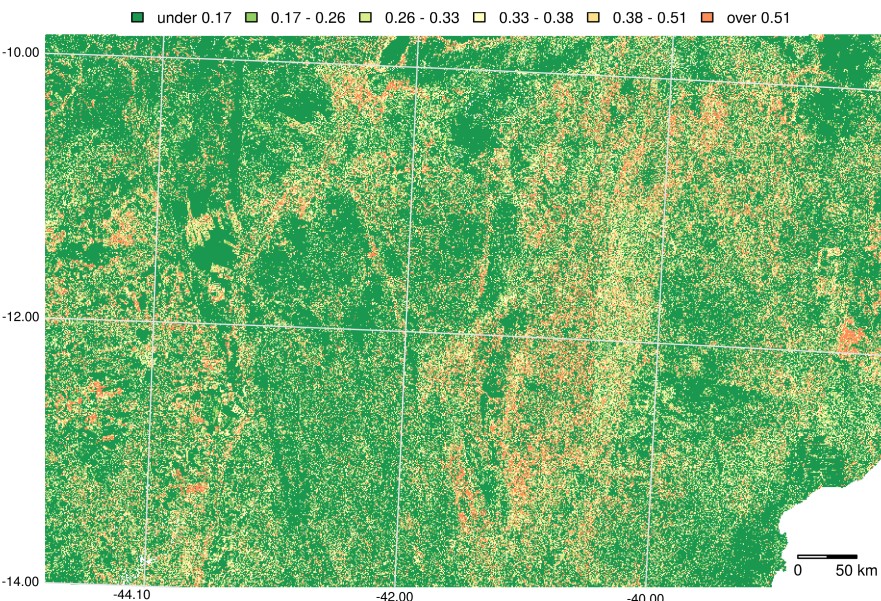

**Figure 6.** *Cont.*

**Turbulence**

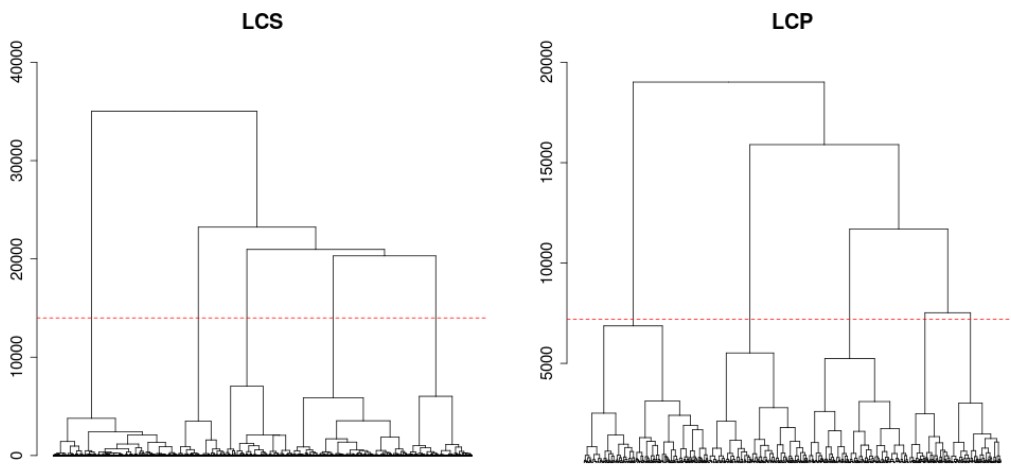

**Figure 6.** Entropy and turbulence index.

## 6.3. Pairwise Dissimilarities between Sequences

Dissimilarities between sequences were calculated using the four different methods (LCS, LCP, OM based on transition rates, and OM based on features) and cluster hierarchical analysis was applied to identify similar sequences. The dendrograms of cluster analysis (Figure 7) suggest five large groups of sequences, except for the clustering based on LCP. However, all the dendrograms were cut at five clusters in order to compare the results derived from the different dissimilarities measures.

**Figure 7.** *Cont.*

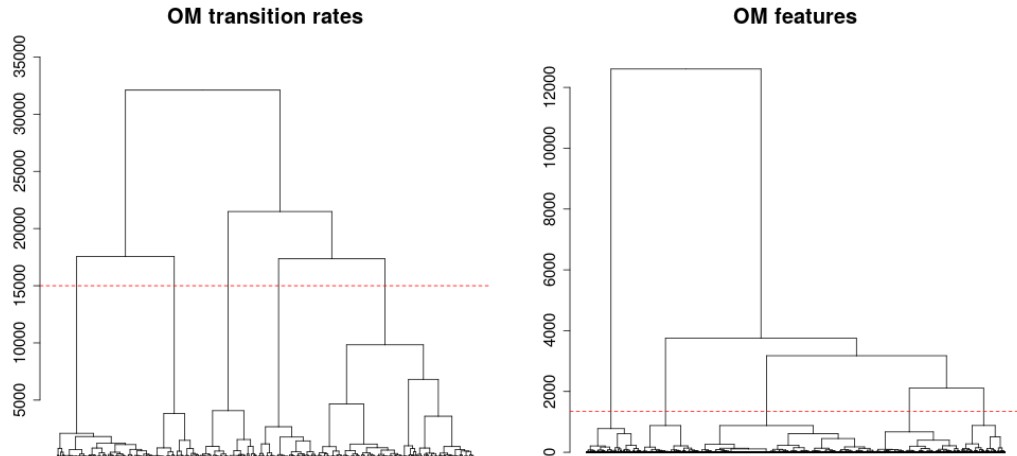

**Figure 7.** Dendrograms of cluster analysis. Longest common subsequence (LCS), longest common prefix (LCP), and optimal matching (OM). The vertical axis refers to the distance between clusters.

In Figure 8, it can be observed that the clustering give different results depending on the dissimilarity method used to calculate the distance between sequences. Clusters based on LCS reflect the dominant land cover over time, while clusters based on LCP depend, as expected, on the initial land cover. In the case of OM based on the transition rates, the indel costs between the categories Savanna, Pasture and Mosaic is low because the rates of transition between these categories is high. As a consequence, there is a cluster (number 4, Figure 8) represented exclusively by these three categories. The clusters based on OM using land cover features tend also to group the categories with similar characteristics. It is worth noting that very different sequences are included into the same cluster (for example, cluster 2 of LCP or 4 of LCS) because the dendrograms were cut at five clusters. Using a larger number of clusters will enable to separate them. The clusters based on different dissimilarity measures exhibit also different spatial distributions (Figure 9).

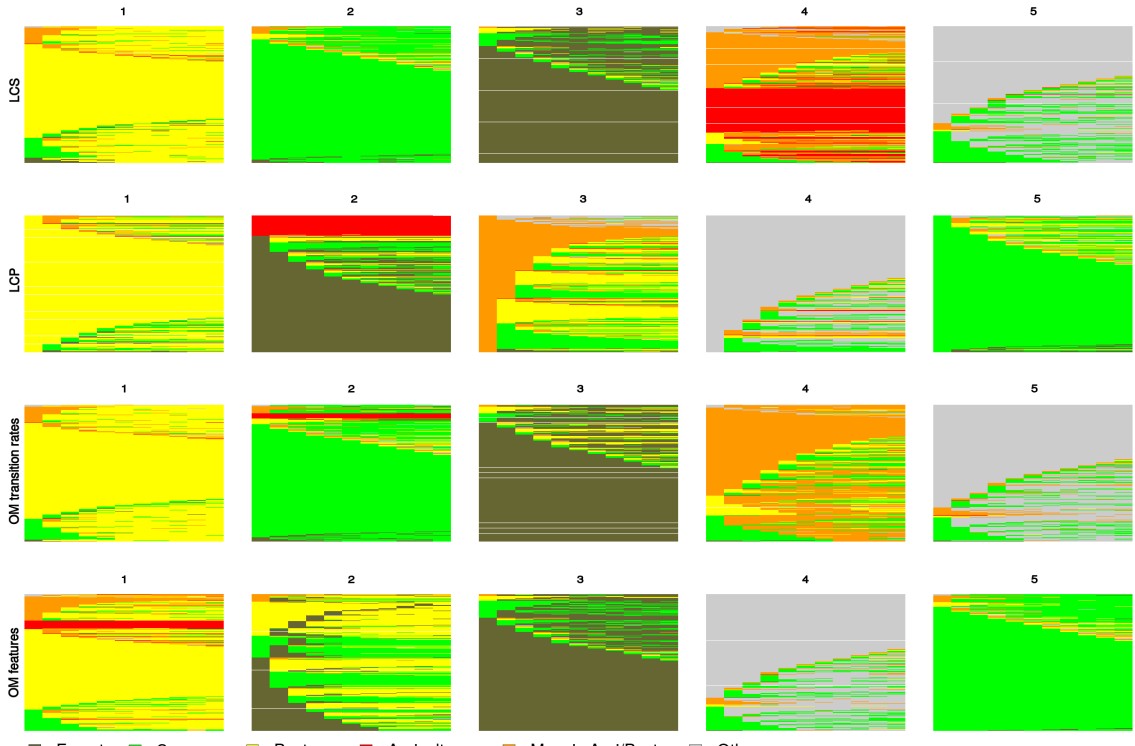

**Figure 8.** Sequences belonging to the different clusters using the four dissimilarity indices.

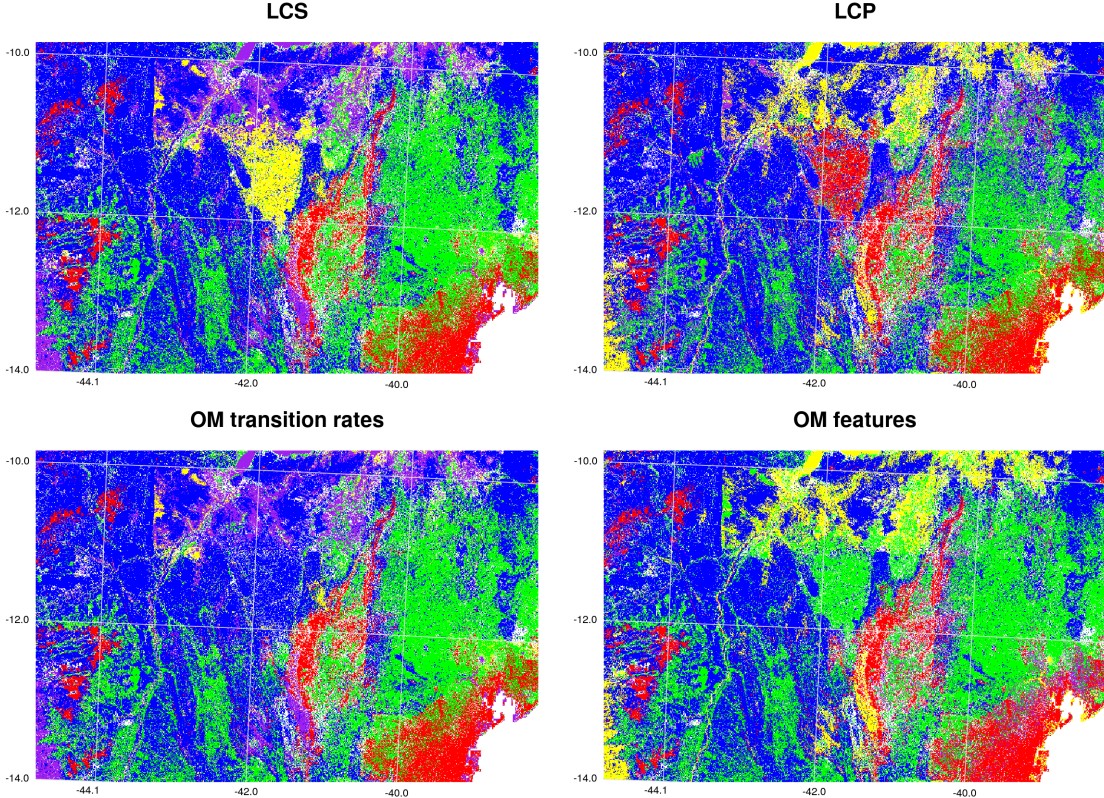

**Figure 9.** Distribution of clusters obtained through the four types of dissimilarities. Colors were assigned to each cluster in order to enhance similarity between maps.

The choice of the method used to calculate the dissimilarities can be based on the patterns of changes considered as more important. For instance, LCP will focus on the state at the beginning of the time series in order to assess the trajectories of sequences sharing the same original land cover. LCS tend to group sequences sharing the same land cover categories over the entire period. OM based on transitions rates will tend to separate sequences which differences concern categories with rare transition. OM based on features enables the user to focus on certain transitions. For instance, savannas shift easily and rapidly into grasslands or woodlands due to anthropogenic (e.g., fire regime), biotic (e.g., woody encroachment), and abiotic factors (e.g., change in rainfalls) [43]. Thus, the land categories are not totally exclusive since a complex and dynamic land management occurs in the study area (e.g., pasture, agriculture, mosaic of pasture and agriculture categories are not completely exclusive). This spatio-temporal complexity leads to a large number of sequences with frequent changes between these three categories. Classification errors due to spectral confusion can also produce erroneous changes in the sequences. OM based on the transition rates can eventually enhance a rare transition which is in fact an error. OM based on features can allow to deal with classification errors and ambiguities through the choice of the indel cost values. However, the understanding of the algorithms and the rationale behind them is not straightforward. For instance, according to Studer and Ritschard [30], the OM distance, which is based on the longest partially matched subsequence, depends on the "common backbone" between trajectories. Nevertheless, this measure has been criticized due to the lack of sociological meaning of the substitutions and indel operations, and their related costs. In the LUCC implementation, the same questions remain.

### 6.4. Assessment of the Effect of Covariates

The effect of covariates on the observed sequences was examined using the dissimilarity between sequences based on OM using the land cover features. Table 3 shows that the covariates have an effect on the observed sequences, as all the covariates have a significant contribution ($p = 0.02$) to explain the dissimilarity between sequences, although the relationship is weak (Pseudo $R^2$ below 0.05).

As a following step, the effect of covariates on deforestation was evaluated. For this, the analysis was restricted to sequences that exhibit the subsequences "Forest to Forest", "Forest to Pasture", "Forest to Mosaic" and "Forest to Agriculture". Figure 10 shows the most common subsequences. They subsequences shows complex interchanges between forest, pasture lands and mosaics of pasture and agriculture. Then, the subsequences that discriminate significantly areas near (distance lower than one kilometer) and far away from roads (distance larger than one kilometer) were determined using a chi-square test. In the Figure 11, the color of each bar depends on the Pearson residual of the Chi-square test. For residuals below $-2$ (red), the subsequence is significantly less frequent than expected under the independence assumption, while for residuals greater than 2 (dark blue), the subsequence is significantly more frequent. It can be observed that, at distance superior to one kilometer, the subsequences of interchange between forest, savanna and pasture are less frequent and at contrary sequences leading to forest recovery and forest permanence are more common. The other covariates have also an effect on the frequency of subsequences, as for instance the slope (Figure 12).

**Table 3.** Multi-factor ANOVA results.

| Variable | PseudoF | Pseudo $R^2$ | $p$ Value |
|---|---|---|---|
| Elevation | 77.31 | 0.013 | 0.02 |
| Slope | 93.51 | 0.015 | 0.02 |
| Distance from roads | 40.25 | 0.007 | 0.02 |
| Annual precipitation | 213.51 | 0.035 | 0.02 |
| Precipitation of driest quarter | 256.69 | 0.042 | 0.02 |
| Total | 210.55 | 0.174 | 0.02 |

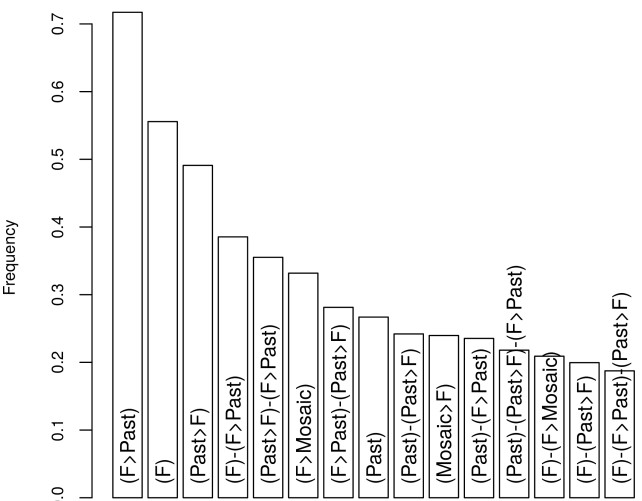

**Figure 10.** Most common subsequences.

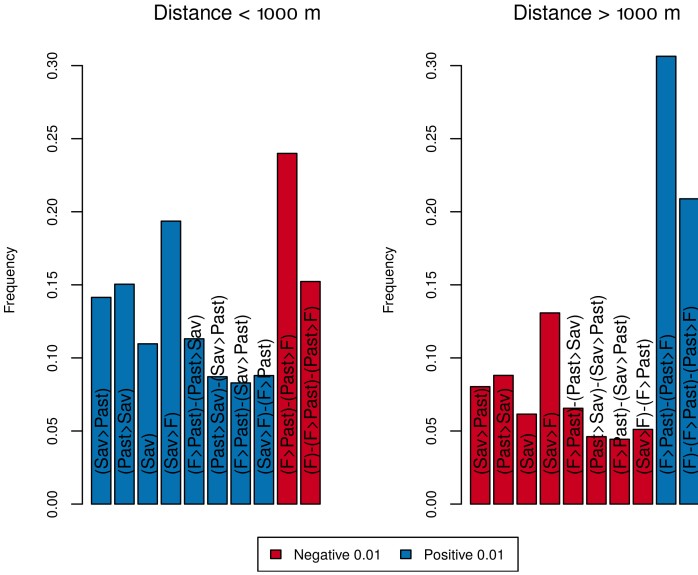

**Figure 11.** Ten most discriminating subsequences between areas proximate to a road (distance < 1000 m) and distant (distance > 1000 m).

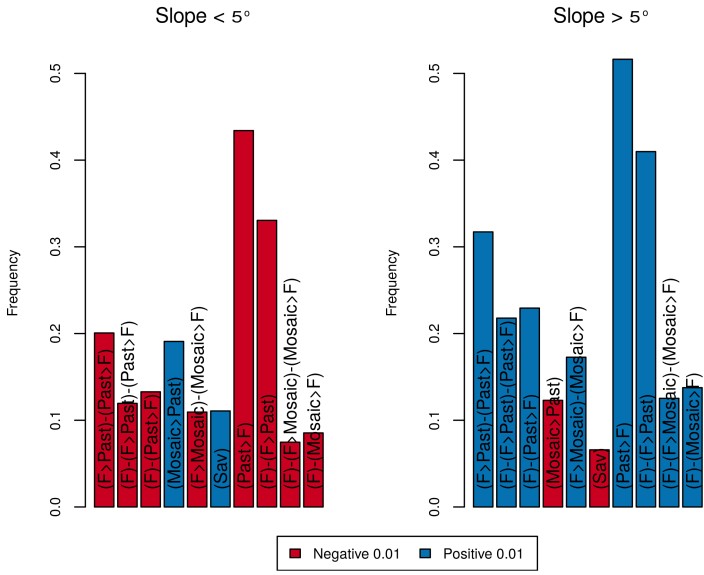

**Figure 12.** Ten most discriminating subsequences between areas in moderate slope (<5°) and steeper slopes (>5°).

*6.5. Limitations and Potentials of Sequence Analysis in Land Change Studies*

The information used to produce the sequences is extremely different from the data commonly used for this type of analysis, which is obtained from interviews or censuses. Based on the maps, it is easy to obtain a very large amount of observations, which exceeds the algorithm and computer capacities. In order to reduce the quantity of data, maps were resampled to 500 m, which can be considered as a systematic sampling. Due to spatial autocorrelation, we consider that only very infrequent sequences can be lost during resampling. Then, an additional random sampling of 46,000 pixels has to be done in order to carry out sequence analysis. Another important difference, is the spatial component of LUCC data. Sequence analysis outputs should be mapped in order to

assess spatial patterns or apply spatially oriented analysis such as local operators (e.g., more frequent sequence based on spatial filtering).

A more serious problem is the accuracy of the maps obtained from remotely sensed images. Due to factors as the difference of acquisition dates, atmospheric conditions, phenology and spectral confusion, the multidate maps present likely errors and incongruencies [44,45]. For instance, under the assumption of a random spatial distribution of error, the probability of obtaining a sequence entirely accurate based on a series of seven maps with overall accuracy of 80% is only 0.21 ($0.8^7$). Due to error propagation, the accuracy decreases rapidly with the increase in the number of dates. The errors likely augment the number of observed sequences with many erroneous variants of a "true" sequence. The temporal filtering (modal category of three year periods) likely reduced some amount of error but could eventually substitute a misclassified category by another wrong category or produce new errors in some cases. The effects of misclassification on the dissimilarity computing and the clustering of sequences need to be examined more deeply.

Despite these limitations, this novel approach can be extremely useful in addressing some of the most pressing questions facing the LUCC research community. Processes of land cover modification such as land use intensification, forest degradation, fragmentation, vegetation recovery constitute the most dominant drivers of biodiversity loss globally, altering the abundance, composition, distribution and functioning of biological diversity [46]. For example, Ochoa-Gaona et al. [47] showed that floristic diversity is related to the frequency and duration of certain land uses. The assessment of most of these processes depends on the temporal periodicity of observations over time. For instance, time series derived from remote sensing data allow for determining cropping cycles, the extent of fallow land, or the frequency of fallow periods [48]. The need for data with a high temporal resolution to coincide with the dynamics of land change patterns suggests the use of monitoring systems with high temporal resolution [41]. High temporal resolution data can also contribute to improving the performance of LUCC prospective models avoiding the errors resulting from considering only two past dates in Markov projections and overcoming the limitations of the assumption of stationarity [49,50].

However, the increasing availability of land cover data with high temporal resolution poses important challenges to its analysis. The LUCC research community should take advantage of the experience in the analysis of sequences from other disciplines and adapt these methods to the concerns of the land change science. Life course trajectory analysis seems particularly promising due to the similarities that exist between the life course sequences and the LUCC sequences (e.g., the importance of the chronology).

## 7. Conclusions

In this work, analysis techniques developed for life course trajectories were applied to sequences of land use/cover obtained from the classification of remote sensing imagery. The sequence analysis allowed the definition of relevant land cover categories and land cover sequences, comparison of sequences, and the search for sequence patterns and related patterns with drivers. Additional analysis can eventually predict future land cover based on the previous sequence and the effects of sequences. However, it is necessary to build theories which try to explain sequences in LUCC behavior.

As pointed by Crews and Young [43], the linkage of pattern to process is a crucial part of the scientific method as it potentially enables (i) developing approaches to prospect possible future outcomes or resiliencies in particular landscapes and (ii) creating theories in order to generalize explanations helpful in other landscapes. The results of the case study show the enormous potential of the methods used in life course trajectory analysis when applied to the analysis of land cover sequences.

**Supplementary Materials:** A R script file along with example data and a guide are available online at http://www.mdpi.com/2073-445X/8/2/30/s1 and https://github.com/jfmas/sequences.

**Author Contributions:** J.-F.M. conceived and designed the paper, analyzed data and wrote the paper; R.N.d.V. and W.F.-R. elaborated MapBiomas cartography and revised the manuscript.

**Acknowledgments:** Cartography was provided by the MapBiomas project (http://mapbiomas.org/). This study was carried out during a sabbatical stay of the first author at *Universidade Federal da Bahia* and *Universidade Estadual de Feira de Santana*. It received support from the National Institute of Science and Technology in Interdisciplinary and Transdisciplinary Studies in Ecology and Evolution (IN-TREE), from the *Coordenação of Aperfeiçoamento de Pessoal de Nível Superior* (CAPES) in the framework of the INCT call—MCTI/CNPq/CAPES/FAPs 16/2014 as well as *Programa de Apoyos para la Superación del Personal Académico* (PASPA) at the *Dirección General Asuntos del Personal Académico* (DGAPA), Universidad Nacional Autónoma de México (UNAM). We thank the two anonymous reviewers for their careful reading of the manuscript and their many insightful comments.

**Conflicts of Interest:** The authors declare no conflict of interest.

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
