# Peer review of "Analysis of High Temporal Resolution Land Use/Land Cover Trajectories"

_land, doi:10.3390/land8020030_

Round 1
Reviewer 1 Report
The paper is well-written and generally easy to follow. Its biggest contribution is the novel application of methods from other disciplines to the Land Change science. The paper at times seems too technical, and little space is devoted to a higher-level discussion of how these methods/approaches can be useful in addressing some of the most pressing questions facing the LUCC research community. I would like to see this addressed in the conclusions section. My detailed comments are below.
Abstract
Line 7 – remove elevation and rainfall because they were not included in this paper
Line 8 – change “interest” to “the potential” or “usefulness”
Key words: remove data mining because you do not discuss this term at all in the paper
Line 27 – remove second “density”
Line 41 – “word” should be “world”
Line 47 – add “course” into “life trajectories”
Line 53 – add more references here – at least one for each of the disciplines – genetics, marketing and clinical studies
Line 55 – associate should be “associating”
Line 68 – remove repeated “biomedical research”
Line 118 – remove the word “section”
Figure 2 – Why doesn’t elevation raster cover the entire study area rectangle? Please revise, so it does. Also, add a scale bar
Line 135 – Add here how many initial land cover classes are in the map, and list them
Line 138 – remove precipitation data form this list because they are not part of the analysis in the paper
Line 149 – Since you used modal category in a 3-year period, were there some pixels that were classified as a different land cover each year (i.e., forest, savanna, pasture)? If so, then how did the algorithm select a modal category? Please explain
Line 151 – resampled “to”
Line 151 – explain how you chose 500m resolution
Line 158 – add an explanation of what it measures and how it can be interpreted
Line 160 – explain what a “subsequence” is and give an example using Figure 1
Line 161 and further – what is the source of all equations? It is not clearly stated. Please be explicit about it in the text
Line 171 – add “The” before “larger the distance”
Line 173 – Is there a formula for LCP? If it is the same as ACS, then you need to state it. The whole paragraph (lines 173-176) is a bit confusing, please revise it
Lines 172 and 176 – the resulting distance is the same. It would be more interesting if you gave two examples with a different resulting distance
Line 207 – “a value…was assigned” – are these indel costs value? If so, please add this to the text
Line 216 – Results and Discussion – I recommend you subdivide this section into the same sections as the Methods (6.1-6.4). It will be easier for the reader to relate each method to the corresponding findings
Line 217 – should be a section 6.1 “Preprocessing”
Line 227 – should be section 6.2 with the same title as section 5.2
Line 229 – where on Figure 5 can this be seen? - “An increase in grassland area can be observed at the expense of the savanna”?
Line 235-236 – it would be useful to add a map of longitudinal entropy to Figure 6, so the reader can see the correlation (r=0.93)
Figure 5 – add labels to the Y-axis
Line 237 – should be section 6.3 with the same title as section 5.3
Lines 327-239 – cluster hierarchical analysis is not mentioned in the Methods section; please add a few sentences in section 5.3
Lines 242-251 – This paragraph describes one of the most important findings of the paper, and I think you need to provide a longer discussion here. For example, you need to explain why very different sequences end up being included into the same cluster (for example, cluster LCS4). It seems contradicting the goal of “cluster hierarchical analysis was applied to identify similar sequences.” (line 239). Is it because you cut the dendrograms at five clusters? What are the implications of this decision for the subsequent analysis?
Also, there are some clusters that are similar regardless of the method used (i.e., LCS2; LCP5; OM-TR2; OM-F5). Explain why it is so and what are the implications? How can this information be used further?
Figure 8 – add legend for colors so it is easier to interpret the figures
Line 252 – section 6.4 would be here, with the same title as 5.4
Line 256-268 – what is the goal of this analysis? Why is it important and how can it be used? Please explain
Figures 10,11 and 12 – label the Y axis
Figure 11 – exclude negative 0.05 and neutral from the legend
Figure 12 - exclude negative 0.05, positive 0.05 and neutral from the legend
Line 269 – please add here more about limitation of this study. For example, uncertainty introduced during your preprocessing (aggregating land cover categories and finding modal category and resampling to 500 m) needs some discussion. Also, discuss the novelty of your approach and what it contributes to the LUCC body of literature
Line 280 – this is an important statement – “can enable users to focus on the land processes considered more important and deal with classification errors and ambiguities” - you need to elaborate more and give some examples
Reviewer 2 Report
This paper implements the life course trajectories to investigate land change through the analysis of sequences of land use/cover. The paper is generally easy to follow, but several issues have be considered before publishing.
The innovation of this paper. Why use the life course trajectories in LULC? The reason for that is not simpy the life course trajectories have been used in the LULC. The authors should explain that from the LULC rather than the method.
The data quality of land cover maps. The maps were classified from 30m remote sensing images. How about the classification accuracy? The change analysis between different classification maps under different years should consider their difference in imaging condition, e.g., illumiation, sun angle, season or others.
The dissimilarity measure problem. Why implments the three measures? Which one is better from the experiment?
The conclusion should be rewritten. The authors should describe the analysis result of life course trajectories in the implemented dataset in the conclusion and abstract part.
Round 2
Reviewer 2 Report
The paper has been greatly improved by the authors. The reviewer thank your careful revision. i think the paper can be published after handling two small issues:
Figure 9 should be centered in the page;
Considering the LULC study in this paper, two popular papers are recommended as the future work for further study.
a) the subpixel method: Quantifying sub-pixel surface water coverage in urban environments using low-albedo fraction from Landsat imagery[J]. Remote Sensing, 2017, 9(5): 428.
b) Designing an experiment to investigate subpixel mapping as an alternative method to obtain land use/land cover maps[J]. Remote sensing, 2016, 8(5): 360.
Author Response
- Size of Figure 9 has been reduced (the figure was not within the margins due to its size)
- Small errors in references were also corrected (formating of book chapters)
- small error in table 1 was corredted (class Mangrove was missing)
- The two references suggested by reviewer 2 have not been included. Both references concerned unmixing and subpixel classification. However the present paper is based on the analysis of classified images (maps) from the MapBiomas project database and does not involve image processing. Moreover, no subpixel analysis was used during image processing in the framework of MapBiomas.
